# Antioxidant and Signaling Role of Plastid-Derived Isoprenoid Quinones and Chromanols

**DOI:** 10.3390/ijms22062950

**Published:** 2021-03-14

**Authors:** Beatrycze Nowicka, Agnieszka Trela-Makowej, Dariusz Latowski, Kazimierz Strzalka, Renata Szymańska

**Affiliations:** 1Department of Plant Physiology and Biochemistry, Faculty of Biochemistry, Biophysics and Biotechnology, Jagiellonian University, Gronostajowa 7, 30-387 Krakow, Poland; beatrycze.nowicka@uj.edu.pl (B.N.); dariusz.latowski@uj.edu.pl (D.L.); kazimierz.strzalka@uj.edu.pl (K.S.); 2Faculty of Physics and Applied Computer Science, AGH University of Science and Technology, Reymonta 19, 30-059 Krakow, Poland; agnieszka.trela@fis.agh.edu.pl; 3Malopolska Centre of Biotechnology, Jagiellonian University, Gronostajowa 7A, 30-387 Krakow, Poland

**Keywords:** antioxidants, lipid peroxidation, oxidative stress, plastochromanol, quinols, reactive oxygen species, signaling pathways, tocochromanols, tocopherols

## Abstract

Plant prenyllipids, especially isoprenoid chromanols and quinols, are very efficient low-molecular-weight lipophilic antioxidants, protecting membranes and storage lipids from reactive oxygen species (ROS). ROS are byproducts of aerobic metabolism that can damage cell components, they are also known to play a role in signaling. Plants are particularly prone to oxidative damage because oxygenic photosynthesis results in O_2_ formation in their green tissues. In addition, the photosynthetic electron transfer chain is an important source of ROS. Therefore, chloroplasts are the main site of ROS generation in plant cells during the light reactions of photosynthesis, and plastidic antioxidants are crucial to prevent oxidative stress, which occurs when plants are exposed to various types of stress factors, both biotic and abiotic. The increase in antioxidant content during stress acclimation is a common phenomenon. In the present review, we describe the mechanisms of ROS (singlet oxygen, superoxide, hydrogen peroxide and hydroxyl radical) production in chloroplasts in general and during exposure to abiotic stress factors, such as high light, low temperature, drought and salinity. We highlight the dual role of their presence: negative (i.e., lipid peroxidation, pigment and protein oxidation) and positive (i.e., contribution in redox-based physiological processes). Then we provide a summary of current knowledge concerning plastidic prenyllipid antioxidants belonging to isoprenoid chromanols and quinols, as well as their structure, occurrence, biosynthesis and function both in ROS detoxification and signaling.

## 1. Introduction

Reactive oxygen species (ROS) comprise both radical and nonradical forms. The radical ROS include superoxide (O_2_^•−^), hydroperoxide (HO_2_^•^), hydroxyl radical (OH^•^) and organic radicals, such as alkoxy and peroxy radicals (RO^•^ and ROO^•^, respectively). The most important nonradical ROS in plants are singlet oxygen (^1^O_2_), hydrogen peroxide (H_2_O_2_) and organic hydroperoxides (ROOH), especially those that are formed during lipid peroxidation. ROS are byproducts of various metabolic pathways. In photosynthetic organisms they are formed as a result of action of various electron-transport chains and some enzymes, but also in a situation when photosynthetic pigments act as photosensitizers [1,2]. ROS effects may be both harmful and beneficial. The first occurs when ROS are generated in high amounts. Such a situation is called oxidative stress and leads to damage of cell components. To protect themselves against excessive ROS, plants have evolved an elaborate system of their detoxification based on antioxidant enzymes and low-molecular-weight antioxidants [3]. The beneficial role of ROS is connected with their role in signal transduction, as well as the antimicrobial activity during pathogen attack. ROS are required for many essential signaling cascades, as well as for the progression of several basic biological processes, i.e., cellular proliferation and differentiation [4,5,6].

Plants are particularly prone to oxidative damage because oxygenic photosynthesis causes an increased O_2_ concentration in chloroplasts. Moreover, the photosynthetic electron transfer chain in thylakoid membranes is an important source of ROS. Therefore, chloroplasts are the main site of ROS generation in plant cells under light. Stress conditions, such as excessive light exposure, cause enhanced ROS formation. In this case, absorbed light energy cannot be efficiently used for photosynthetic reactions, which leads to the increase in rate of unwanted side reactions of O_2_ excitation or partial reduction [6,7].

The important role in protection of plastids against ROS is played by prenyllipid antioxidants belonging to isoprenoid chromanols and quinols, which are able to scavenge oxygen and organic radicals, as well as quench and scavenge ^1^O_2_. These compounds are also crucial for protection of membrane lipids from lipid peroxidation. It is worth mentioning here that thylakoid-membrane fluidity is increased with the presence of lipids with polyunsaturated fatty-acid residues, which are much more prone to oxidation than saturated lipids. Apart from their antioxidant function, prenyllipids are known to play a role in cellular signaling [8,9].

## 2. The Sites of ROS Production in Chloroplasts

The most important types of ROS that are produced in plants during normal metabolic processes are ^1^O_2_, O_2_^•−^, H_2_O_2_ and OH^•^ [7,10]. High reactivity of ROS enables them to react with lipids, proteins, nucleic acids, pigments and carbohydrates, which leads to the disturbance of cell metabolism and destruction of its components. This results in the increase in electrolyte leakage, reduction of photosynthetic yield, acceleration of senescence and cell death [11,12]. To counteract oxidative stress, cells enhance synthesis of low-molecular-weight antioxidants and activity of ROS-detoxifying enzymes [7,13,14]. While the older literature reports pointed mainly to the negative role of ROS in plants, the current opinion also indicates the positive role of ROS as essential pro-life signals participating in redox sensing, signaling and regulation [15,16]. In plant cells, ROS are formed in the majority of cell compartments. The most important sources of ROS are chloroplasts, mitochondria and peroxisomes. Substantial amounts of ROS are also formed in the cell wall and apoplast. This process is usually beneficial, as it enables cross-linking of cell-wall polymers [6,17,18]. The sites of ROS generation in thylakoid membranes and place of action of selected antioxidants are shown in Figure 1.

Superoxide-generating systems include electron leakage from various electron transport chains, plastid terminal oxidase, NADPH oxidases and xanthine oxidase present in peroxisomes. However, the greatest amount of this radical is formed on the PSI acceptor site. The process, called the Mehler reaction, comprises electron leakage from Fe–S clusters of PSI and from reduced ferredoxin, resulting in the O_2_ reduction [1,5,10,11]. It has been suggested that the phylloquinone acceptor (A_1_) is the major contributor to O_2_^•−^ generation inside the thylakoid membrane in vivo [19] (Figure 1). In plastids, low amounts of O_2_^•−^ are also produced within the PQ pool and as result of electron leakage from PSII [1,20]. The enhanced reduction of molecular oxygen to O_2_^•−^ takes place when the plant is exposed to high light and/or when the dark phase of photosynthesis is slowed down due to decreased CO_2_ availability or disturbed activity of Calvin-cycle enzymes [1,5].

The superoxide anion undergoes rapid dismutation to H_2_O_2_, a relatively stable type of ROS able to diffuse on long distances [7]. The enzymes catalyzing the above-mentioned reaction are superoxide dismutases. There are various types of these enzymes differing with the metal ion in their active center. Plastids of higher plants contain membrane-bound Cu/Zn dismutase and Fe-containing dismutase (the presence of the latter in plastids of some plant species has been questioned and needs to be re-evaluated) [21,22]. Detoxification of H_2_O_2_ in the chloroplasts is catalyzed by ascorbate peroxidases (APXs), which reduce this ROS to H_2_O using ascorbic acid (Asc) as an electron donor. This mechanism, called the water–water cycle, is a very efficient method of excessive-energy dissipation [23]. APXs have high affinity to their substrate. In plastids, both thylakoid membrane-bound and soluble forms of these enzymes are found [24]. Plant chloroplasts do not contain catalase (CAT), therefore APX is considered as the main H_2_O_2_ detoxifying enzyme in this cell compartment [25].

The most important source of H_2_O_2_ in plants are peroxisomes, where this type of ROS is produced by glycolate oxidase, an enzyme participating in photorespiration, as well as during β-oxidation of fatty acids [26]. Considering chloroplasts, Pospisil [20] suggested that H_2_O_2_ might be formed at the donor side of PSII, in the process of water biophotooxidation. However, this reaction is unlikely to occur in nature. It is more probable that H_2_O_2_ is formed on the acceptor side of PSII, as a result of the reaction of O_2_ with plastoquinol (PQH_2_) in the situation when the PQ pool is highly reduced [20].

Singlet oxygen is thought to be the major ROS responsible for light-induced loss of PSII activity, and it was also shown to be responsible for over 80% of nonenzymatic lipid peroxidation in leaves [23,27]. This type of ROS is formed as a result of excitation energy transfer from the triplet state of excited chlorophylls (^3^Chl *) on triplet oxygen. Light absorption by chlorophyll results in the formation of the singlet excited state (^1^Chl *). However, if the excited energy of the singlet state is not transferred to the reaction center nor used for photosynthetic reaction of electron transfer, ^1^Chl * may undergo intersystem crossing leading to ^3^Chl * and subsequent reacting with ^3^O_2_ generate ^1^O_2_ [28,29]. The most important site of such photosensitized ^1^O_2_ formation is P680 in PSII. This process is enhanced when the photosynthetic electron transfer chain is over-reduced, and therefore the electron transfer from ^1^P680 * to pheophytin is hindered. In addition, in such a situation there is also a risk of charge recombination between oxidized P680 and reduced pheophytin, which results in the formation of ^3^P680 * [12,17,27,30]. Due to its high reactivity, ^1^O_2_ easily oxidizes PSII core peptides, especially peptide D_1_, leading to PSII photoinhibition [30]. Oxidized D_1_ peptide needs to be replaced with the newly synthesized D_1_ in a process called D_1_ peptide turnover [30,31].

In the in vitro experiments, ^1^O_2_ was shown to be produced in light-harvesting complexes [27]. In addition, Suh et al. [32] observed ^1^O_2_ formation in an illuminated isolated cytochrome *b_6_f* complex. The exact mechanism has not been elucidated. The authors postulated the role of the Fe–S cluster in this process, while the others suggested participation of the chlorophyll *a* molecule bound to the cytochrome complex [32,33]. Singlet oxygen may be also produced during plant greening, when etioplasts change into chloroplasts and thylakoid membranes are formed. These rearrangements pose a threat of occurrence of disorganized chlorophyll, which can act as photosensitizer [34]. Protochlorophyllide, free chlorophyll and chlorophyll-degradation products are also capable of photosensitization [10,35]. To prevent this, plants synthesize chlorophyll binding proteins including ELIPs (early light-induced proteins) and WSCPs (water-soluble chlorophyll proteins) [36].

The possible diffusion distance of ^1^O_2_ was calculated to be ~10 nm, therefore this ROS is able to leave the site of its formation and react with thylakoid membrane lipids, pigments and proteins [30,31,37]. Singlet oxygen can be quenched by carotenoids, both present in PSII and freely diffusing in thylakoid membranes [38,39]. It can be also quenched and scavenged by prenyllipids and hydrophilic low-molecular-weight antioxidants [24,38,40] (Figure 1).

## 3. Structure, Occurrence and Biosynthesis of Isoprenoid Quinones and Chromanols

Among the roles played by prenyllipids in plants, their antioxidant action is an important one. The major prenyllipid antioxidants belong to the groups of isoprenoid chromanols, isoprenoid quinols and carotenoids. Isoprenoid chromanols and quinols, as well as xanthophylls, which are oxygen-containing carotenoids, are amphipathic compounds. Their molecules are composed of a polar head-group(s) (a chromanol or quinol ring in the case of isoprenoid chromanols and quinols, respectively, and two terminal rings in the case of xanthophylls) and an apolar prenyl side-chain, which anchors them in lipid bilayers [9,41]. Due to above-mentioned properties, these compounds are crucial for antioxidant protection of membranes and lipid-storage sites [42].

The most important and most common isoprenoid chromanols are tocopherols (Toc) and tocotrienols (T_3_). These compounds are also essential nutrients for animals and humans [14]. The isoprenoid side-chain of Toc is fully saturated and derived from phytol diphosphate, while the T_3_ side-chain is unsaturated and derived from geranylgeranyl diphosphate [41]. The chromanol rings of Toc and T_3_ vary with the number and position of methyl substituents, therefore we distinguish four vitamers: α, β, γ and δ. There are also other, rarely occurring compounds related to Toc and T_3_. The known examples are tocomonoenols (T_1_) and tocodienols (T_2_), differing in the degree of side-chain desaturation; desmethyl and didesmethyltocotrienol, lacking methyl groups in the chromanol ring when compared to δ-tocotrienol; as well as Toc derivatives such as tocopherol acids, tocopherol phosphates and tocopherol esters [43]. Isoprenoid chromanols with longer side-chains are also known: plastochromanol-8 (PC-8) and its oxidized derivative, hydroxy-plastochromanol (PC-OH) [44].

Naturally occurring Toc have three asymmetric carbon atoms, i.e., C2 of the chromanol ring and C4′and C8′ of the side-chain, all of them in configuration *R*. The double bonds in the T_3_ side-chain are in all-trans configuration, while the configuration of the C2 atom in the ring is the same as in Toc [41]. The structures of isoprenoid chromanols are shown in Figure 2. All isoprenoid chromanols mentioned above are antioxidants, but among them, α-Toc is the most important in biological systems [45]. PC-8 was suggested to be very effective in detoxification of ROS and organic radicals in the hydrophobic core of lipid bilayer due to its longer side-chain [46]. This compound was shown to play a role in ^1^O_2_ detoxification in *Arabidopsis thaliana* (*Arabidopsis*) [47].

The most important plastidic isoprenoid quinol is plastoquinol-9 (PQH_2_-9). This compound, together with its oxidized form, plastoquinone-9 (PQ-9), which is also called PQ A, are crucial elements of the photosynthetic electron transfer chain in all organisms performing oxygenic photosynthesis [9]. Similar, but rarely occurring compounds also have been discovered, among them plastoquinones with shorter side chains (like PQ-8, -7, -4, phytylplastoquinone) or differing in the number of methyl groups in the quinone ring (such as 3-demethylplastoquinone-9 or 6-methyl-dihydrophytylplastoquinone, which has also more saturated side-chain). The hydroxy derivative of PQ-9, PQ C and its ester PQ B also have been described in the literature [43]. Recently, it was discovered that PQ-9 hydroxy-derivatives containing more −OH groups in the side-chain may be produced under high light stress [48]. The strong antioxidant action of PQH_2_-9 has been observed both in vitro and in vivo. Interestingly, PQ also displays some antioxidant properties [46,49]. Apart from PQ-related compounds, the oxidation product of α-Toc, called α-tocopherol quinone (α-TQ), and its reduced form, α-tocopherol quinol (α-TQH_2_), are known to occur in plant chloroplasts and to play an antioxidant role in these organelles [42]. Phylloquinone, the main function of which is transferring electrons inside PSI, has been also postulated to occur in membranes and participate in ROS detoxification; however, this hypothesis appeared in one paper and was not confirmed later by other scientists [9]. Recently, this compound also was reported to be present in plastoglobules, which are thought to be its reservoir [50]. In the liposome model system, phylloquinone was shown to detoxify ^1^O_2_ [46]. Mitochondrial respiratory quinone, UQ and its reduced form ubiquinol (UQH_2_) also display antioxidant properties [42]. The dominating UQ types in higher plants are UQ-9 and UQ-10 [43]. Ubiquinone antioxidant function in plants has not been examined as widely, as in the case of Toc or PQH_2_; however, there are literature data concerning this topic [51]. The structures of chosen isoprenoid quinones are shown in Figure 3.

Isoprenoid chromanols occur commonly in higher plants and in the majority of cyanobacteria [52]. They are also present in green, red and brown algae, as well as in diatoms [53,54,55,56]. Considering subcellular localization, these compounds can be found mainly in chloroplasts, both in membrane fractions and in plastoglobules, which are plastid lipid storage sites [14,52,57]. The other types of plastids, such as leucoplasts and chromoplasts, also contain isoprenoid chromanols. The presence of these compounds in other compartments, such as mitochondria, vacuoles, microsomal fractions and the nucleus, has been reported in the literature, although there were doubts concerning purity of the fractions used for analyses [14]. In the seed cells, isoprenoid chromanols are stored in leucoplasts and oleosomes. The latter may contain up to 40% of the total seed chromanol pool [58].

In higher plants, isoprenoid chromanols occur in all tissues, but the highest amounts are present in green leaves and seeds. The main sources of Toc are plant-derived oils, seeds, fruits and vegetables [14,58] (Table 1). In green parts of the plants, the predominant Toc form is α-Toc. It can be also found in nonphotosynthetic organs (roots, tubers, flowers, fruits, bulbs, seeds), but usually its content is lower [41]. γ-Tocopherol (γ-Toc) occurs in leaves, usually in minor amounts when compared to α-Toc; however, it dominates in seeds of many species. The other forms of Toc, β- and δ-, occur in minor amounts in seeds. There are also exceptions. In leaves of Patagonian plant *Misodendrum punctulatum*, the dominant Toc form is γ-Toc. This species also contains significant amounts of δ-Toc [59]. Moreover, γ- and δ-Toc were found in cuticular waxes of some species [14]. Tocotrienols are present in the seeds of some species, primarily belonging to monocots. In the endosperm of cereal grains, they can constitute up to 50% of the total chromanol pool [41]. These chromanols were also found in some fruits and in the latex of rubber tree [60]. The occurrence of rare isoprenoid chromanols, such as T_1_, T_2_ or desmethyltocotrienols, is limited to certain species and organs, where these compounds can be found in minor amounts. For example, α-T_1_ and α-T_2_ were found in palm oil, δ-T_1_ in kiwi fruit, γ-T_1_ in etiolated seedlings of *Phaseolus coccineus*, γ-T_2_ in pumpkin seed oil, all three T_1_ mentioned above in *Kalanchoe daigremontiana* leaves, and desmethyl and didesmethyltocotrienols in rice bran [43]. Tocochromanol acids, phosphates, esters and other derivatives were found in leaves, bark and seeds of some species [43]. Plastochromanol-8 occurs in smaller amounts in the leaves of many species, and it can be also found in seeds [44], while PC-OH has been detected in *Arabidopsis* leaves [57,61]. The isoprenoid chromanol content depends on the species, tissue, organ, stage of development and environmental factors [41].

Being an oxidation product of α-Toc, α-TQ is present in minor amounts in green and nongreen plant tissues. It was observed to accumulate during senescence of leaves and chromoplast development [9].

Due to its role in oxygenic photosynthesis, PQ/PQH_2_-9 needs to be present in chloroplasts and in cyanobacteria. In chloroplasts, PQ-9 occurs in thylakoids and plastoglobules; the latter are the storage site for lipids in these organelles. This compound is also present in the inner chloroplast envelope where it is synthesized. The occurrence of PQ-9 is not limited to green tissues; minor amounts of this prenyllipid were also found in flowers, fruits, roots, bulbs and etiolated leaves [9]. Plastoquinones with shorter side-chains were found in some species, e.g., PQ4 in the autumn leaves of *Aesculus hippocastanum* as a minor component; PQ-3 in spinach chloroplasts; PQ-8 in summer leaves of *A. hippocastanum*, maize and *Ficus elastica*; and phytylplastoquinone in protozoan *Euglena gracilis* strain Z. Demethylplastoquinones occur in *Iris hollandica* bulbs, while 6-methyl dihydrophytylplastoquinone occurs in *Costus speciosus* seeds [43]. PQ C, being a product of PQ side-chain oxidation, and PQ B accumulate in older leaves [42].

In the biosynthesis of all isoprenoid chromanols and quinones, the head-group and side-chain precursors are synthesized separately, then specific prenyltransferase catalyzes their condensation, and the product of this reaction undergoes further modifications [9,60]. The biosynthesis of Toc, T_3_, T_1_, PC-8 and PQ in higher plants is already well known. The synthesis of UQ and phylloquinone also has been elucidated; however, this is beyond the scope of present paper. The biosynthetic pathways of isoprenoid chromanols and PQ evolved in cyanobacteria, but not all of these organisms are able to produce Toc. As cyanobacteria are the ancestors of plastids, the ability for Toc and PQ synthesis was acquired by photosynthetic eukaryotes [9,41]. However, there is an interesting difference. In higher plants, the Toc and PQ biosynthetic pathways share a head-group precursor, homogentisate (HGA) (Figure 4), while cyanobacteria have separate routes for synthesis of these compounds. In the cyanobacterial PQ biosynthetic pathway, the quinone ring is derived from *p*-hydroxybenzoate, which is also a head-group precursor for UQ biosynthesis [62]. In addition, genes encoding prenyltransferase and some enzymes responsible for further modifications of the head-group are homologues of proteobacterial UQ-biosynthetic genes [63,64].

Much of the research necessary for elucidation of the Toc biosynthetic pathway in higher plants has been carried out on *Arabidopsis*. Certain mutants defective in vitamin E biosynthesis and therefore called *vte* mutants turned out very useful for pathway unraveling. The group of such mutants was obtained and numbered (from 1–6), and later the responsible genes were identified and consecutively named *VTE1*, *VTE2*, etc. The exact role of certain products of these genes was determined later (see below in the text and in Figure 4).

In higher plants, HGA is synthesized from tyrosine-derived catabolite *p*-hydroxyphenylpyruvate (HPP) [65]. In most plant species, tyrosine is synthesized in plastids via the arogenate pathway, although some species also have an alternative, nonplastidic route, called the *p*-hydroxyphenylpyruvate pathway [66]. The enzyme responsible for conversion of HPP into HGA, *p*-hydroxyphenylpyruvate dioxygenase (HPPD), occurs in chloroplasts or cytosol, depending on the plant species [67].

The isoprenoid side-chain precursor is formed via multistep condensation of five-carbon compounds, dimethylallyl pyrophosphate (DMAPP) and isopentenyl pyrophosphate (IPP). There are two main pathways for the synthesis of both DMAPP and IPP, the mevalonate (MVA) pathway and the 1-deoxy-D-xylulose-5-phosphate (DXP) pathway (also called the methylerythritol phosphate pathway) [68]. The alternative MVA pathway, differing from the formerly known one in its last two steps, also has been described in the literature, although its occurrence is limited to *Arachaea* and some bacteria [69]. The DXP pathway is present in cyanobacteria and in the plastids of higher plants. The latter group also has an MVA pathway, which enzymes are localized in cytosol [65]. Both pathways are present in the majority of algal clades, but some green and red algae contain only DXP pathway [70]. It was postulated that the DXP pathway serves as the main source of precursors for the synthesis of plastidic isoprenoids. However, it was observed that at least in the case of overproduction of DMAPP and IPP in the cytoplasm, they can be imported to the chloroplast [67]. The condensation of DMAPP and IPP into longer chains is catalyzed by enzymes belonging to the class of polyprenyl pyrophosphate synthases. The length of the final product depends on the type of synthase involved. Geranylgeranyl pyrophosphate (GGPP) synthesized by the GGPP synthase is a precursor for the T_3_ side-chain and a substrate for geranylgeranyl reductase, which reduces GGPP to phytol pyrophosphate (PPP) needed for Toc synthesis [8]. Ten genes of GGPP synthases have been identified in the *Arabidopsis* genome. The products of seven of these genes are most probably localized in plastids, among them the main paralogue responsible for the formation of GGPP used for Toc and T_3_ synthesis [71]. In higher plants, phytol derived from chlorophyll degradation can be phosphorylated to PPP. The enzymes responsible for these reactions are the phytol kinase VTE5 and the phytyl phosphate kinase VTE6. Nowadays, it is thought that side-chain precursors for Toc biosynthesis in the leaves are obtained mostly via phytol recycling [67]. The side-chain precursor for T_1_ biosynthesis is an intermediate of the GGPP to PPP reduction [72,73]. The precursor of the PQ nonaprenyl side-chain is synthesized by solanesyl-diphosphate synthase (SPS). The genome of *Arabidopsis* contains two SPS genes, *AtSPS1* and *AtSPS2*. *AtSPS2* is targeted to the plastids and it is thought to be the main enzyme responsible for biosynthesis of the PQ side-chain [74].

Condensation of precursors is the first step specific to Toc, T_1_, T_3_ and PQ biosynthetic pathways. It is catalyzed by the enzymes belonging to the UbiA family, which differ with the specificity for the isoprenoid substrate. Homogentisate phytyltransferase (HPT, encoded by the *VTE2* gene) participates in the synthesis of Toc and T_1_; homogentisate geranylgeranyl transferase (HGGT) is necessary for T_3_ biosynthesis; while homogentisate solanesyl transferase (HST) plays a role in PQ formation [73].

In the biosynthesis of Toc, T_3_ and T_1_, further reactions of head-group modifications are carried out by the same set of enzymes, which do not discriminate substrates varying in the saturation of the side-chain [73]. The product of above-mentioned condensation of the head-group and side-chain precursors can either undergo cyclization or methylation (Figure 4). The cyclization, carried out by the tocopherol cyclase (TC, encoded by the *VTE1* gene) leads to the formation of a chromanol ring of Toc, T_1_ or T_3_ δ-forms. However, in green tissues of higher plants and in the majority of plant seeds, the condensation of precursors is followed by methylation, driven by 2-methyl-6-phytyl-1,4-benzoquinone methyltransferase (MPBQ MT, encoded by *VTE3*). The resulting compound containing two methyl groups in its quinone ring is then a substrate for TC, in this case synthesizing γ-forms of Toc, T_1_ or T_3_. Both δ- and γ-tocochromanol forms are substrates for γ-tocopherol methyltransferase (γ-TMT, encoded by *VTE4*), and the products of these reactions are β- and α-tocochromanols, respectively [60].

The biosynthesis of PQ requires fewer steps (Figure 4). The head and tail condensation product, 2-methyl-6-solanesyl-1,4-benzoquinone, is methylated, which leads to PQH_2_-9 formation. This reaction is catalyzed by the same enzyme that participates in tocochromanol synthesis, an MPBQ MT. Plastoquinol-9 may be then oxidized to PQ-9 enzymatically or nonenzymatically. Alternatively, it can be a substrate for TC, which converts it to PC-8 [52,73].

Among tocochromanol biosynthetic enzymes, HPT, MPBQ MT and γ-TMT occur in chloroplast envelope, whereas TC is present in plastoglobules [41]. Considering prenyltransferase necessary for PQ biosynthesis, it was shown that in *A. thaliana*, HST localizes in the chloroplast, most probably in its envelope membrane [51].

The genes encoding enzymes responsible for tocochromanol biosynthesis are known, which enables genetic manipulations [45]. There were attempts to perform genetic engineering of the tocochromanol biosynthetic pathways, aimed at obtaining transgenic lines with an increased level of these compounds or the increased amount of α-Toc, which is preferentially absorbed by humans [75,76,77,78].

## 4. Antioxidant Function of Prenyllipids

The antioxidant properties of isoprenoid chromanols have been widely examined. The majority of the research was focused on α-Toc due to its common occurrence in plants and importance for human nutrition. In the in vitro and in vivo experiments, it was shown that α-Toc efficiently scavenges O_2_^•^^−^ and lipid radicals (such as L^•^, LO^•^ and LOO^•^), and also quenches and scavenges ^1^O_2_ [42,79]. This makes α-Toc important for membrane protection, where it inhibits lipid peroxidation, both radical and ^1^O_2_-mediated [12,42,80]. The antioxidant properties of other Toc vitamers, T_3_ and PC-8, also have been examined, and they turned out to be qualitatively similar to those of α-Toc [46,81]. Toc are also known to protect thiol groups of proteins [82]. In the reaction of radical scavenging, isoprenoid chromanols donate a hydrogen atom from the hydroxyl group in their chromanol ring. The resulting radical is relatively stable due to electron delocalization in the aromatic ring [41,60]. It was shown in vitro that the α-tocopheroxyl radical resulting from radical scavenging by α-Toc can be reduced back to Toc by Asc, PQH_2_, UQH_2_ or phenolic compounds [60]. It seems plausible that Asc and PQH_2_ play the most important role in re-reduction of the tocopheroxyl radical in plastids. Chloroplasts contain the extensive enzymatic system responsible for Asc recycling, whereas PQ is reduced to PQH_2_ by the photosynthetic or chlororespiratory electron transfer chain [24,83,84]. The reaction of α-Toc with ^1^O_2_ leads to the formation of unstable 8a-hydroperoxy-α-tocopherone. This compound can be re-reduced to α-Toc by Asc or further oxidized to stable form, α-TQ [60]. α-Tocopherol quinone may be enzymatically reduced to α-TQH_2_ in an NADPH-dependent reaction [85].

Kobayashi and DellaPenna [86] reported that α-TQ and α-TQH_2_ were the only stable α-Toc oxidation products found in *Arabidopsis* leaves. They also postulated the presence of a plastid-based enzymatic cycle enabling recycling of α-TQH_2_ into α-Toc [86]. It was shown that α-TQ accumulates in leaves of runner bean exposed to low temperature and high light, and that the level of this compound is higher in evergreen plants [87].

Tocopherols, together with β-carotene and PQH_2_, were postulated to quench ^1^O_2_ in the PSII [40]. The turnover of D_1_ peptide was shown to be correlated with the turnover of α-Toc in *Chlamydomonas reinhardtii* (*C. reinhardtii*) [85]. It was shown that ^1^O_2_ inhibits translation of the D_1_ peptide, therefore it was suggested that Toc rather protects D_1_ resynthesis than directly act as ^1^O_2_ quencher in PSII [88,89].

In spite of the potent antioxidant properties of α-Toc, the experiments on mutants with impaired biosynthesis of this compound showed that most often there are no phenotypic differences between the wild type and mutants, both in unstressed and stressed conditions [52,90,91,92]. This led scientists to conclude that there is a redundancy between cellular antioxidants. The decrease in Toc content was usually accompanied by an increase in the amount of other antioxidants [82]. The more enhanced increase in Asc and total thiol content, as well as in superoxide dismutase and APX activity, was observed in *C. reinhardtii* grown in the presence of toxic concentrations of heavy metal ions and an inhibitor of HPPD when compared to the heavy metal treated algae in the absence of inhibitor of Toc and PQH_2_ synthesis [55]. Deficiency of Toc in *Arabidopsis* results in accumulation of carotenoid zeaxanthin and vice versa [90,93]. Double *Arabidopsis* mutant *vte1npq1*, lacking both Toc and zeaxanthin, displays strong photosensitivity and ^1^O_2_-mediated lipid peroxidation under high light conditions [90]. The overlapping protective functions of carotenoids and Toc were also reported for *C. reinhardtii* [94]. Considering other antioxidants, the double *Arabidopsis* mutant *vte1cad2*, with impaired Toc and glutathione synthesis, was more sensitive to photo-oxidative stress than single mutants *vte1* and *cad2* [95]. In the case of *Arabidopsis* mutants with impaired late steps of α-Toc biosynthesis, it was shown that the Toc biosynthetic intermediates, 2-methyl-6-phytyl-1,4-benzoquinone and 2,3-dimethyl-6-phytyl-1,4-benzoquinone (DMPBQ), also display antioxidant properties. These compounds were accumulated in the *vte1* mutant of *Arabidopsis* [52,57].

The accumulation of tocochromanols in seeds suggests that these compounds fulfill a special role there. Tocochromanols seem to be crucial for protecting seed-storage lipids from peroxidation [8,96]. *Arabidopsis* mutant *vte2* produces seeds that have greatly reduced longevity in comparison to the wild type [97]. What is more, this mutant displays disturbed seedling development, i.e., impaired cotyledon expansion, limited root growth, defects in storage-lipid metabolism and a dramatic increase in nonenzymatic lipid peroxidation [97,98]. However, such a seedling phenotype was not observed in *vte1* mutants accumulating DMPBQ instead of Toc [99]. Analysis of gene expression during seed germination of *vte2* and *vte1* mutants showed that in *vte1* seeds, 12 genes were upregulated when compared to the control, while in *vte2*, the number of such genes exceeded 160. The difference in gene expression between the wild type and the *vte1* mutant indicates that not all functions of Toc in germinating seeds can be fulfilled by DMPBQ [98]. Interestingly, it has been shown recently that seed longevity is associated with a high proportion of γ- and δ-tocochromanol homologues, rather than total tocochromanol content, in several rice cultivars [100,101].

Recently, vom Dorp et al. [96] reported on a long-missing enzyme, a phytyl-phosphate kinase (VTE6), which links phytol from chlorophyll degradation with Toc synthesis. The *VTE6*-overexpression line showed increased levels of phytyl-diphosphate and Toc in seeds. The corresponding *Arabidopsis* mutants were Toc-deficient in leaves and exhibited a severe reduction in seed longevity and plant growth [96]. However, it was speculated that impaired growth of the *vte6* mutants is not caused primarily by Toc deficiency, but might rather be connected with free phytol accumulation [96].

In recent years, it was clearly recognized that isoprenoid quinols including PQH_2_ have potent antioxidant properties. Similarly to isoprenoid chromanols, they are able to scavenge inorganic (O_2_^•−^, perferryl radical) and lipid radicals, as well as quench and scavenge ^1^O_2_ [9,42,46,80,81]. In organic solvents, PQH_2_ was shown to scavenge H_2_O_2_ [102]. Isoprenoid quinols also play a role in Toc recycling [44]. Radical scavenging by isoprenoid quinols leads to the formation of semiquinone radicals, which disproportionate to quinones and quinols [103]. Isoprenoid quinones have antioxidant properties, i.e., they are able to scavenge ^1^O_2_ and O_2_^•−^, but they are less effective than reduced forms [44,46,81]. Apart from the hydrophilic head-group, a long, unsaturated chain of PQ may also participate in ^1^O_2_ scavenging. The product of this reaction is PQ C. Similarly, the oxidation of the side-chain of PC-8 by ^1^O_2_ results in the formation of PC-OH. Both these compounds were detected in vivo [104]. Recently, Ferretti et al. [48] presented direct evidence for chemical quenching of ^1^O_2_ by PQH_2_-9. In their study on the *Arabidopsis* line, the overexpression of the *solanesyl diphosphate synthase 1* gene, involved in PQH_2_-9 biosynthesis, resulted in the decrease of ^1^O_2_ formation when compared to the wild type, and accumulation of PQH_2_-9 oxidized forms, PQ and various hydroxy-plastoquinones [48]. It was also shown that overexpression of the PQ-9 biosynthetic genes provides an enhanced tolerance of *Arabidopsis* plants to strong excess light, decreasing the leaf bleaching, lipid peroxidation and PSII photoinhibition [105]. Partial inhibition of PQH_2_ and α-Toc biosynthesis in *C. reinhardtii* led to increased O_2_^•−^ formation when compared to the control [55].

## 5. Role of Prenyllipid Antioxidants in Abiotic Stress Response

Oxidative stress commonly occurs when plants are exposed to various abiotic stress factors, such as high light, heat, cold, drought, heavy metals, salinity or flood. The enhanced ROS production is a consequence of the disturbance of metabolism in stressed plants [106,107,108]. Considering the signaling role of ROS, they are known to participate in regulation of various developmental processes, such as the cell cycle, growth or programmed cell death, but also are crucial for signal transduction during response to abiotic and biotic stress [109,110,111]. Research aiming at identification of chemical nature and cellular source of ROS formed during response and acclimation to various stress factors is being carried out [112]. Usually the researchers expose plants to one type of stress, but one has to remember that in the field, plants are subjected to many stress factors at the same time, and the growth conditions are different than controlled and relatively stable laboratory conditions [113]. Semchuk et al. [114] observed that Toc content in field-grown *Arabidopsis* was 8- to 12-fold higher than in laboratory-grown-plants.

Excessive light causes over-reduction of the photosynthetic electron transfer chain and, as a consequence, enhanced ROS formation in plastids. The major type of ROS formed in such conditions is ^1^O_2_, which causes PSII photoinhibition and oxidizes lipids with polyunsaturated fatty-acid chains [23]. Another ROS formed during high light exposure is O_2_^•−^, able to initiate radical-mediated lipid peroxidation [38]. Plants have evolved various strategies to protect themselves from excessive light, includingsynthesis of low-molecular-weight antioxidants, and production of antioxidant enzymes is one of them [41,93].

The participation of tocochromanols in response to high light has been widely examined. It was shown that tolerant species increase tocochromanol levels under stress, while sensitive ones show a loss of vitamin E, which leads to oxidative damage and cell death. The changes in Toc content depend also on stress intensity. α-Tocopherol has been the most studied prenyllipid antioxidant in the aspect of its role in response to high light because it is the main Toc vitamer in green leaves [115]. However, the protective function of other isoprenoid chromanols also has been reported.

Szymanska and Kruk [52] have shown for the first time that in addition to α-Toc, PQH_2_ plays a crucial role in high light acclimation in plants. Experiments on *Arabidopsis* mutants with disturbed Toc biosynthesis led authors to observe that in high light-exposed plants PQH_2_ content increased 10-fold independently from the Toc level. The increase in PQH_2_ was mainly attributed to the photochemically inactive fraction of the PQ-pool localized in plastoglobules [52]. The important role of PQH_2_ in response to high light stress was also reported for green microalga *C. reinhardtii* [49]. The increase in α-Toc and PQH_2_ content was also observed in plants exposed to high light in combination with low temperature and nutrient deficiency [116].

Tocochromanols are also important in plant acclimation to low- and high-temperature stresses. A recent analysis revealed a correlation of the increase in α-Toc and PQ level with the normal functioning of the photosynthetic apparatus under high-temperature stress in tomato leaves [117]. An increased level of α-Toc was found in several species of mosses growing in harsh Antarctic conditions [118]. Wang et al. [119] reported that *SGD1*, a rice homologue of the *VTE2* gene, is essential for cold tolerance in this cereal. *Sgd1-2* mutants lacked Toc, displayed significantly dwarf and small-grain phenotypes and were hypersensitive to cold stress [119].

In response to water deficiency, plants close their stomata to limit transpiration, which results in a decrease in CO_2_ availability. In such a situation, the reactions of the dark phase of photosynthesis slow down, and the photosynthetic electron transfer chain becomes over-reduced. As a consequence, ROS formation in chloroplasts is enhanced [120]. The increase in α-Toc content in plants exposed to drought was observed in *Arabidopsis*, tobacco, field-grown rosemary, holm oak, sage, lemon balm and strawberry leaves [115,121,122]. More recently, it was shown that in maize (*Zea mays* L.), the PC-8 level increased in response to reiterated drought stress. Plastochromanol-8 contents paralleled those of vitamin E, particularly α-Toc, which suggests that PC-8 may help Toc to protect photosynthetic apparatus from damage [123].

As the induction of oxidative stress is also a significant consequence of heavy metal toxicity, prenyllipid antioxidants are thought to participate in the response to heavy metal-induced stress. Toc content was shown to increase in heavy metal-exposed plants [124]. In response to exposure to copper and cadmium ions, the content of α-Toc in *Arabidopsis* leaves increased 6- and 5-fold, respectively [125]. The upregulation of expression of genes encoding enzymes participating in Toc synthesis was also observed: copper induced expression of the *VTE2* gene, while cadmium induced *HPPD* and *VTE2* (at the first stage of stress), and *VTE5* (at a later stage) [125]. An increase in the content of prenyllipids was observed in *C. reinhardtii* exposed to Cr_2_O_7_^2−^, Cd^2+^ (α- and γ-Toc and PQ + PQH_2_) and Cu^2+^ (only Toc) for two weeks [126].

Another stress factor known to have an impact on Toc level is salinity. In *Arabidopsis*, salt treatment leads to an increase in α-Toc and PC-8 by 50 and 60%, respectively [127]. Total Toc content, mainly represented by vitamer α, increased 4-fold in salt-treated wild-type tobacco. Transgenic lines with a silenced *VTE4* gene instead of α-Toc accumulated γ-Toc [128]. An increase in the Toc level in salt-exposed plants was also observed in basil [129].

The effects of individual stress factors are intensively studied, but there are also works that investigate combinations of different stress factors. For example, through the use of the *vte5* mutant, which has a decreased expression of phytol kinase, and therefore a reduced level of α-Toc, the crucial role of α-Toc in the adaptation of tomato to combined high temperature (HT) and high light (HL) stress has been demonstrated. Interestingly, *vte5* mutants were highly sensitive to combined HT + HL, but not to the single stresses [130].

## 6. Role of ROS and Prenyllipid Antioxidants in Cellular Signaling

The subtle balance between ROS production and scavenging is sensed by the cell. The large network of genes which expression changes in response to ROS is called the “ROS gene network” [131]. In model plant *Arabidopsis*, more than 152 genes were identified that belong to this network and participate in the regulation of ROS metabolism in higher plants [132]. The oxidative stress often occurs as a result of the exposure to different stress factors, both abiotic and biotic. Aerobic organisms therefore evolved signaling pathways in which ROS are involved in transduction of signals, leading to the induction of stress responses [133]. In higher plants, ROS also play a role in the regulation of developmental processes [134]. The ROS signaling occurs both inside the cells and on an intercellular level. Plants are able to sense ROS levels and the redox status of their cells [133,135]. The research on lack-of-function mutants and transgenic plants let scientists characterize regulatory proteins involved in ROS-sensing and regulation of ROS-induced gene expression [2,134]. ROS-detoxifying enzymes and low-molecular-weight antioxidants, among them prenyllipids, modulate ROS levels in cells, thus they can indirectly influence ROS-signaling in response to abiotic and biotic stresses [2]. It was proposed that the role of Toc in plant cellular signaling is based on altering ROS levels, rather than on direct regulation of gene expression. On the other hand, the redox state of the PQ/PQH_2_ pool is an important factor that triggers signaling associated mostly, but not only, with the regulation of photosynthesis [42,51]. Figure 5 shows schematic summary of tocochromanols and PQ/PQH_2_ pool regulatory mechanisms.

The participation of Toc in signal transduction has been proposed in the literature. Munne-Bosch et al. [136] observed that α-Toc deficiency in *Arabidopsis* results in reduction of plant growth, increase in anthocyanin content and transient increase in jasmonic acid (JA) content. Due to its antioxidant properties, α-Toc controls the extent of lipid peroxidation and LOOH content in chloroplasts, which may have an impact on JA level and therefore may indirectly influence JA-dependent gene expression [136]. Experiments concerning the participation of Toc in signaling connected with biotic stress response have been carried out recently [137]. The tolerance to bacterial (*Pseudomonas syringae*) and fungal (*Botrytis cinerea*) pathogens was assessed in *Arabidopsis* wild type and two mutants, Toc-lacking mutant *vte1* and mutant *vte4*, which accumulates γ-Toc instead of α-Toc in the leaves. The response of the wild type and mutants was similar in the case of bacterial infections, but a reduced resistance of the mutants was observed in the case of the fungal pathogen [137]. It was shown that mutants infected by fungi displayed a changed lipid profile and delayed induction of JA biosynthesis when compared to the infected wild type. In addition, this effect was accompanied by a reduced level of cytokinin (zeatin) in *B. cinerea*-infected mutants [137].

Expression of genes involved in ethylene biosynthesis, perception and signaling in *Arabidopsis* exposed to salt or water stress strongly varied between the wild type and *vte4* mutants, but not as much as between *vte1* and control [138]. This effect was most evident under salt stress in mature leaves. Mutant *vte4*, accumulating γ-Toc in its leaves, displayed elevated levels of transcripts connected with the ethylene signaling pathway (mainly *ctr1*, *ein2*, *ein3* and *erf1*). In addition, in mature leaves of the *vte4* mutant exposed to salt stress, there was no accumulation of JA, whereas such a response was observed for the *vte1* mutant and the wild type. The authors suggested that various Toc homologues play specific roles in plants [138]. Interestingly, in the experiment on CAM plant *Aptenia cordifolia*, it was observed that a water deficit specifically induced γ-Toc accumulation in leaves [120]. It was also suggested that γ-Toc can interact with signal transduction pathways in which nitric oxide (NO) acts as a messenger, because this form reacts with nitrogen oxides [139].

Preincubation of a tobacco BY2 suspension cell culture with 50 µM α-Toc inhibited activation of mitogen-activated protein kinase (MAPK) after elicitor treatment. This effect was not due to direct inhibition of MAPK, therefore it was concluded that α-Toc acts upstream in the signaling pathway, leading to MAPK activation [140]. The *Arabidopsis adh2-1* mutant with low-Tyr phenotype displayed reduced miRNAs levels, and this effect was reverted by homogentisate supplementation. The miRNA content was also decreased in *vte1* and *vte2-1* mutants, therefore it was suggested that Toc plays a role in miRNAs biogenesis [141].

Tocopherols are also involved in the regulation of plant carbohydrate transport [142]. The *sdx1* (*sucrose export defective 1*) mutant of maize displays disturbances in differentiation of plasmodesmata and bundle-sheath cells [91]. Characteristic phenotypic features of this mutant are defects in photoassimilate transport, death of vascular parenchyma cells, occlusion of plasmodesmata, callose deposition and lack of symplastic connection between bundle-sheath cells and vascular parenchyma [143]. Further research revealed that *sxd1* gene is a homologue of *Arabidopsis VTE1* [142]. A similar phenotype was observed in transgenic potato, in which *StSXD1* expression was silenced using the RNAi method [143], and in Toc-deficient *Arabidopsis* mutants *vte1* and *vte2*, but in that case only when plants were grown at low temperature [144]. The analysis of gene expression in the *vte2* mutant under low-temperature stress has shown that genes associated with solute transport were repressed, while those involved in cell-wall modifications, including genes belonging to the callose synthase family, were induced [145]. Interestingly, introduction of a mutation into the *GSL5* gene, which encodes a major enzyme responsible for pathogen-induced callose deposition, led to the reduction of callose deposition, but did not reverse the photoassimilate export phenotype in double mutant *gsl5vte2* exposed to cold stress [145].

Disrupted photoassimilate transport is not the only effect occurring in *Arabidopsis* mutants of Toc-biosynthetic genes under low-temperature treatment. A changed lipid profile (lower content of linolenic acid and higher level of linoleic acid) was observed in the mutants when compared to the cold-exposed wild type [144,146]. It was shown that different lipid composition resulted from the reduced conversion of dienoic to trienoic fatty acids in the endoplasmic reticulum (ER). Toc was therefore postulated to modulate the polyunsaturated fatty-acid metabolism in this compartment during low-temperature-stress response [146]. The cold-induced phenotype was abolished in double *Arabidopsis* mutants *vte2 fad2* or *vte2 fad6*. *FAD2* and *FAD6* encode fatty-acid desaturases, which affect the conversion of monoenoic to dienoic fatty acids [146]. The suppression of the cold-induced phenotype of the *vte2* mutant was also observed in plants carrying mutations in the TGD 1, 2, 3 and 4 transporters responsible for lipid transport from ER to plastids [147]. The null mutations of genes involved in plastidic pathways of prenyllipid synthesis, *VTE1*, *VTE4* and *LUT1* (encoding an enzyme playing a role in carotenoid biosynthesis), were complemented by the expression of the above-mentioned genes modified so that their products were targeted to the ER. It was therefore postulated that there is exchange of lipids between the ER and chloroplast envelope [148]. This would enable Toc to directly affect processes occurring in the ER.

The signaling role of PQ has been examined more widely than that of Toc [9,42,149]. In experiments aimed to assess the impact of the redox state of the PQ pool on signaling, photosystems were selectively excited or specific inhibitors were used to block PQ reduction or PQH_2_ oxidation [9]. It has been postulated that the redox state of the PQ pool is the major redox sensor in chloroplasts responsible for initiation of many physiological responses to changes of the environmental conditions, especially those related to quality and intensity of light [9].

The plastoquinone-induced signal plays a role both in short- and long-term responses during photoacclimation processes [42]. The well-known example of the first type is connected with the relocation of LHC antennae proteins, called state transitions. In state 1, LHCII antennae are associated with PSII. The low redox state of the PQ pool is responsible for activation of the *b_6_f* complex-dependent protein kinase (STN7 in *Arabidopsis*, STT7 in *C. reinhardtii*), which phosphorylates LHCII. Phosphorylated antennae dissociate from PS II and associate to PSI. When this process is finished, the photosynthetic apparatus is in state 2 [150,151,152]. In higher plants, TAP38 phosphatase, which is thought to be constitutively active, is responsible for LHCII dephosphorylation and a return to state 1 [153]. In higher plants, the migrating antennae comprise about 10–20% of the LHCII pool, and the role of state transitions is to balance energy distribution in photosynthetic apparatus. In *C. reinhardtii*, up to 80% of LHCII dissociate from PS II, but it has been proposed that a large fraction of these antennae stays unbound to photosystems [150,152]. The key function of state transitions in *C. reinhardtii* is the regulation of linear electron transfer and cyclic electron flow (CEF) around PSI. The transition from state 1 to state 2 leads to the enhancement of CEF, which would be beneficial in the situation of increased ATP demand or limited CO_2_ assimilation [153].

The long-term acclimatory response is connected with the regulation of gene expression, which leads, for example, to the adjustment of PSI and PSII stoichiometry in response to changed light conditions. The redox state of the PQ pool regulates the expression of chloroplast genes, including *psaAB* and *psbA* encoding core peptides of photosystems, and *rbcL* encoding a large subunit of rubisco. It is also known to regulate expression of nuclear genes: i.e., *lhcb* encoding antenna complexes, plastocyanin, ferredoxin, nitrate reductase, superoxide dismutase, ascorbate peroxidase and some enzymes of the carotenoid biosynthetic pathway [9,42,154,155,156,157]. The above-mentioned STN7 kinase, as well as chloroplast sensor kinase (CSK) participate in PQ-dependent redox signaling, leading to changes in the expression of plastid genes during acclimation to long-term light-quality shifts [156]. The other elements of this signaling cascade are plastid transcription kinase (PTK) and bacterial-type RNA polymerase sigma factor-1 (SIG-1) [158]. The microarray analysis of gene expression in *Arabidopsis* led to a conclusion that among 663 genes that are differentially expressed depending on light conditions, the expression of 50 of them is regulated by the redox state of the PQ pool [159]. A similar number of genes has been observed to undergo PQ-dependent regulation in other experiments [156]. Petrillo et al. [160] observed that the redox state of PQ pool triggers a signal leading to promotion or inhibition of alternative splicing of selected nuclear genes. The chloroplastic redox-dependent protein kinases were postulated to participate in signal transduction. The stimuli acting in leaves were able to trigger changes in gene splicing in roots [160].

The over-reduction of the PQ pool was demonstrated to induce stomatal closure in *Arabidopsis* [161]. The involvement of the PQ pool redox state signaling in the response to pathogens has also been proposed. It was shown that hypersensitive response to *B. cinerea* in *Mesembryanthemum*
*crystallinum* was accelerated when the PQ pool was in the reduced state. In addition, the redox state of the PQ pool had an impact on changes of catalase and superoxide dismutase activities during pathogen response [162].

The role of PQ in signaling can be also connected with its interaction with ROS. The ROS signaling can be modulated both by scavenging and quenching of ROS by PQ/PQH_2_, but also by participation of an over-reduced PQ pool in ROS generation, which may lead, for example, to the induction of cell death [149,163].

## 7. Conclusions and Future Prospects

The antioxidant functions of Toc are well known nowadays. Numerous in vitro experiments enabled the determination of rate constants for the reactions of Toc and other tocochromanols with different ROS and lipid radicals, as well as identification of their products. The Toc, T_3_, PQ and PC-8 biosynthetic pathways in plants have been elucidated. Toc biosynthesis in cyanobacteria and T_1_ pathway in plants are also known. The identification of tocochromanol biosynthetic enzymes made it possible to obtain transgenic plants and mutants with enhanced or impaired Toc biosynthesis. The mutants let us examine tocochromanol functions in vivo, whereas transgenic plants with increased Toc content would be beneficial in agriculture. However, such lines have not been introduced into commercial breeding yet.

Not all members of above-mentioned groups were equally examined. For example, PC-8 has not been a subject of research as intensive as in the case of Toc. The antioxidant properties of PQ are known to a lesser extent than the participation of this compound in electron transfer. There are also recently discovered, less common chromanols that definitely need further study. Some enzymes connected with Toc metabolism, such as Toc oxidase present in *Phaseolus* young seedlings, are still poorly known. The PQ biosynthetic pathway in cyanobacteria, which differs from that of higher plants, is a subject of ongoing research.

The role of prenyllipid antioxidants in plant stress response has been widely examined. Most research in this field was focused on high light and temperature stress; therefore, some other stress factors, such as various environmental pollutants, need more detailed study.

Determining the role of tocochromanols and plastoquinone in signaling is an important direction of research. The impact of ROS scavenging by these compounds on ROS-mediated signaling is one of the interesting topics that need deeper understanding. The reliable and specific methods for ROS determination inside living cells have to be improved to provide better tools for examining ROS signaling in vivo. The role of prenyllipids in signaling, independent from their reactions with ROS, also needs to be further examined. In particular, the molecular mechanisms of the Toc-mediated signaling cascade are not entirely discovered.

## Figures and Tables

**Figure 1 ijms-22-02950-f001:**
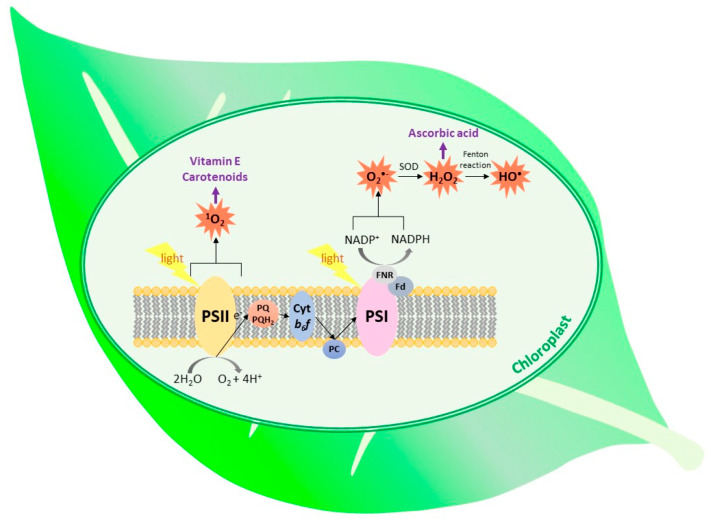
The types of ROS generated within the thylakoid membrane and place of action of the selected antioxidants. (SOD, superoxide dismutase; Fd, ferredoxin; FNR, ferredoxin–NADP^+^ reductase; PC, plastocyanin; cyt *b_6_f*, cytochrome *b_6_f*).

**Figure 2 ijms-22-02950-f002:**
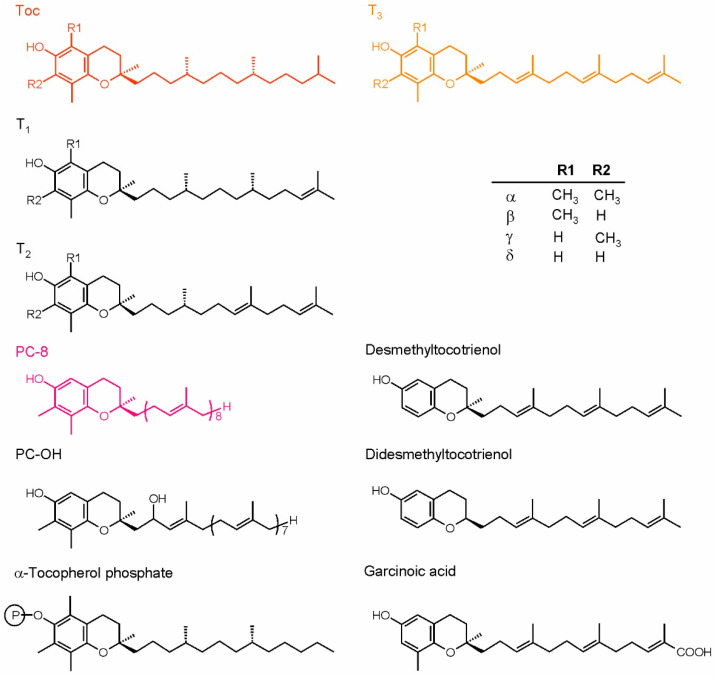
Plant isoprenoid chromanols. The more common compounds are shown in color, while the examples of rare forms are shown in black. The table depicts the substituents configuration in α-, β-, γ- and δ-forms of tocopherols and tocoenols. PC-8, plastochromanol; PC-OH, hydroxy-plastochromanol; T_1_, tocomonoenols; T_2_, tocodienols; T_3_, tocotrienols.

**Figure 3 ijms-22-02950-f003:**
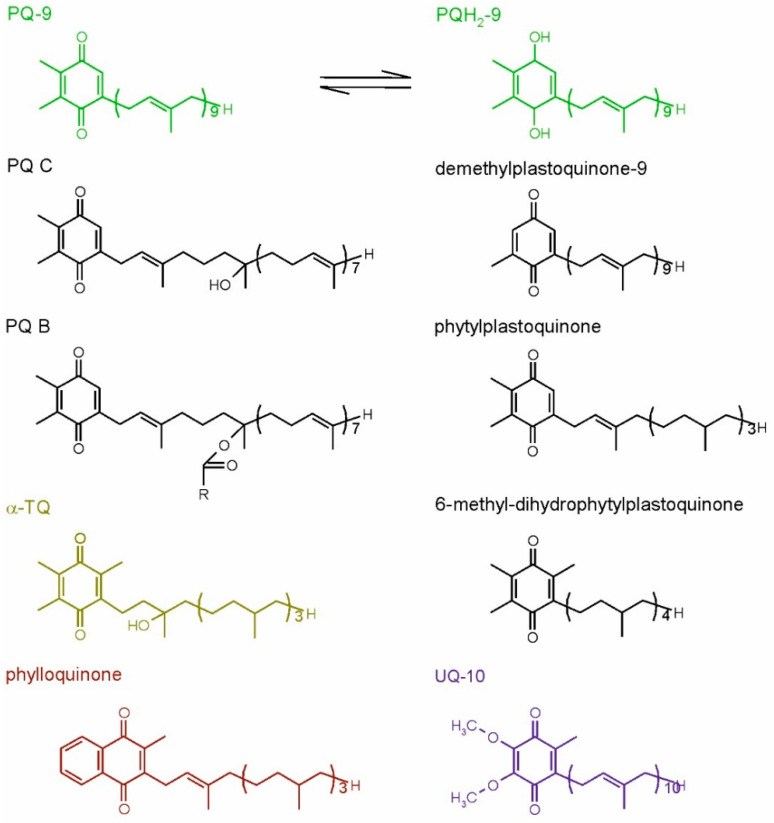
Plant isoprenoid quinones. The common compounds are shown in color, while the examples of rare forms are shown in black. For plastoquinone-9, its reduced, quinol form also is shown. PQ-9, plastoquinone-9; PQ B, plastoquinone B; PQ C, plastoquinone C; PQH_2_-9, plastoquinol-9; R, fatty acid hydrocarbon tail; α-TQ, α-tocopherol quinone; UQ-10, ubiquinone-10.

**Figure 4 ijms-22-02950-f004:**
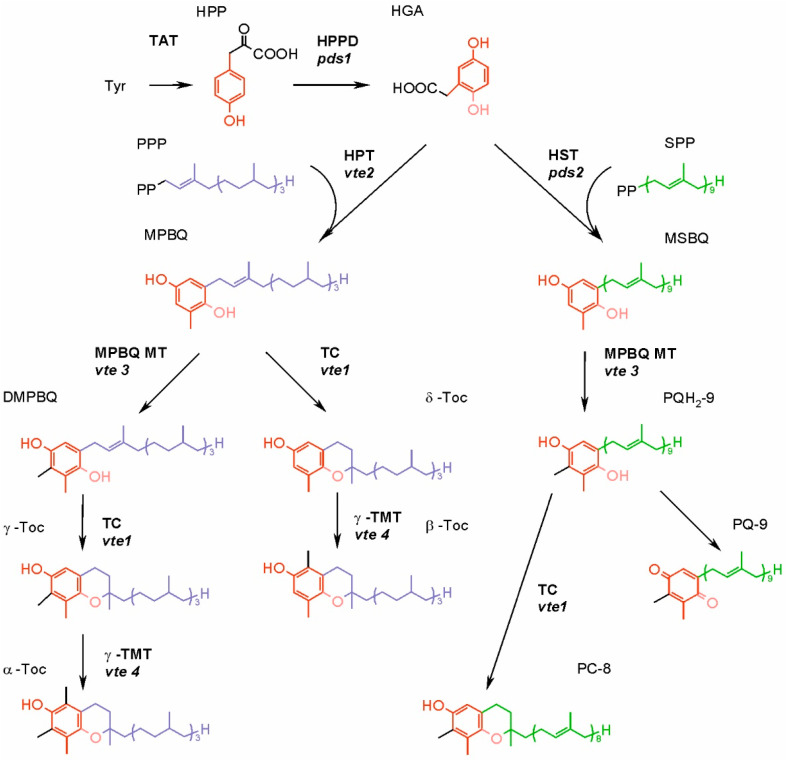
Synthesis of tocopherols, plastoquinone and plastochromanol-8 in higher plants. The mutants of genes encoding enzymes of *Arabidopsis* are given in italics. DMPBQ, 2,3-dimethyl-6-phytyl-1,4-benzoquinone; HGA, homogentisic acid; HPP, *p*-hydroxyphenylpyruvate; HPPD, *p*-hydroxyphenylpyruvate dioxygenase; HPT, homogentisate phytyltransferase; HST, homogentisate solanesyl transferase; MPBQ, 2-methyl-6-phytyl-1,4-benzoquinone; MPBQ MT, 2-methyl-6-phytyl-1,4-benzoquinone methyltransferase; MSBQ, 2-methyl-6-solanesyl-1,4-benzoquinone; PC-8, plastochromanol-8; PQ-9, plastoquinone-9; PQH_2_-9, plastoquinol-9; PPP, phytyl pyrophosphate; SPP, solanesyl pyrophosphate; TAT, tyrosine aminotransferase; TC, tocopherol cyclase; γ-TMT, γ-tocopherol methyltransferase; α-Toc, α-tocopherol; β-Toc, β-tocopherol; γ-Toc, γ-tocopherol; δ-Toc, δ-tocopherol; Tyr, tyrosine.

**Figure 5 ijms-22-02950-f005:**
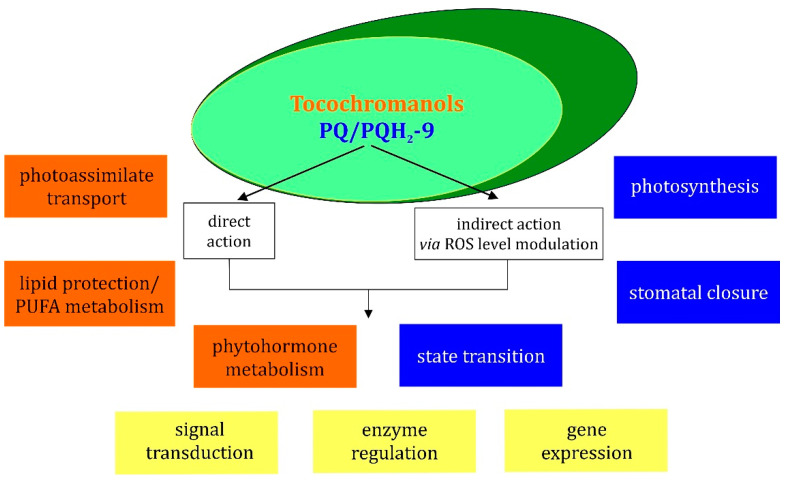
Scheme illustrating the functions of tocochromanols and PQ/PQH_2_-9 in the regulation of different cellular processes. Orange boxes concern tocochromanol functions, blue boxes concern PQ/PQH_2_ functions and yellow boxes concern processes in which both tocochromanols and PQ/PQH_2,_ are engaged.

**Table 1 ijms-22-02950-t001:** The content of isoprenoid tocochromanols in plant organs. Toc, tocopherol; T_3_, tocotrienol [38,60].

Plant Species	Total Chromanol Content [μg/g FW]	Major Homologue (% of Total Chromanols)
Leaves		
*Arabidopsis thaliana*	10–20	90% α-Toc
Parsley	48.1	98% α-Toc
Spinach	30	63% α-Toc
Tobacco	182.2	99% α-Toc
*Ficus elastica*	304.4	99% α-Toc
*Viscum album*	57.7	97% α-Toc
*Betula verrucosa*	307	95% α-Toc
*Populus tremula*	656	97% α-Toc
*Pinus sylvestris*	119	99% α-Toc
Seeds		
*Arabidopsis thaliana*	200–300	95% γ-Toc
Almond	263	97% α-Toc
Corn	60	75% γ-Toc
Flax	236	84% γ-Toc
Rice	17	30% α-T_3_
Wheat	50	56% β-T_3_
Fruits		
Apple	1–4	90% α-Toc
Avocado	15–31	84% α-Toc
Banana	1.5	87% α-Toc
Tomatoes	6.8	78% α-Toc
Cucumber	1.6	50% α-T_3_
Kiwi	14.5	90% α-Toc
Raspberries	37	40% γ-Toc
Strawberries	4.1	68% α-Toc
Other		
Carrot roots	8.7	99% α-Toc
Potato tubers	0.7	90% α-Toc
Sweet potato tubers	3.6	70% α-Toc

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
