# Peer review of "Antioxidant and Signaling Role of Plastid-Derived Isoprenoid Quinones and Chromanols"

_ijms, 2021, doi:10.3390/ijms22062950_

Round 1

Reviewer 1 Report

The review by Nowicka and coll. Titled: “Antioxidant and signaling role of plastid-derived isoprenoid quinones and chromanols” is a tremendous effort to gather an invaluable compendium of useful information on the topic of prenyllipids, diversity synthesis accumulation and functions starting from the different origins of ROS acting as stress molecules during the light reactions of photosynthesis. The review should be published.

I have though a couple of suggestions to improve on the readability of the review and a few minor points.

  1. I think it would be wonderful to provide a schematic illustration on the last part of the review concerning the genetic actors and the diverse signaling pathway to support the thorough written descriptions. This would help making connections between the different elements of literature and probably grasp the complexity of the topic at a glance. Figure 1 is such an attempt that is unfortunately a little to disconnected from the text.
  2. It would also be nice to provide a list of abbreviations and acronyms that are numerous in the paper.

Minor points and suggestions

Abstract:

Line 20: “under light conditions” could be replaced by “during the light reactions of photosynthesis”.

Lines 21 and 22 could be connected since the authors suggest that the two observations are concomitant.

Line 28: for time concordance I would use “then we provide…”.

Main text:

Line 62: maybe rephrase to avoid finalism in the sentence: fluidity is increased with the presence of …

Line 103: the absence of Fe-SOD in some plastids of higher plants seems a little controversial; maybe they were simply not observed or detected.

Pages 9-10: It would be nice to provide the reader with a small introduction on the class of the vte mutants and give its meaning “vitamin E deficient” in the main text as many of them appear in the figure 4.

Line 652 the authors are mentioning the long-term acclimatory response and the role of STN7 in the change of gene expression controlled by the redox state of the PQ pool without mentioning the state transition, which is also controlled by the redox state of the PQ pool acting through STN7. It might be intentional but the author’s reasoning should be given.

Suggestion: “What is more” is often used at the beginning of a sentence that is not always an emphasis of the previous one and could therefore be replace by “In addition” or “moreover”.

End.

Author Response

Dear Reviewer,

Thank you so much for your valuable comments. Please find enclosed the revised version of our review article entitled ‘Antioxidant and signaling role of plastid-derived isoprenoid quinones and chromanols’ (ID ijms-1135554) corrected according to your suggestions.  All changes were provided in the text and were marked in red.

As regards to the specific points:

  1. Additional figure (Fig. 5) was provide as a summarizing illustration on the last part of the review concerning diverse signalling pathways. Indeed, Figure 1 was clarified and modified.
  2. Abbreviations list were added at the end of manuscript.
  3. Corrections within the abstract part (lines 20, 21, 22, 28) were made and marked in red.
  4. Corrections within the main text (line 62, 103) were made and marked in red in the text.
  5. Line 103, the statement concerning the absence of Fe-SOD in plastid of higher plant was changed and highlight in the text.
  6. Page 9-10: small introduction concerning the meaning of “vitamin E deficient” was added.
  7. Line 652: The state transition was and is described in details in above paragraph (lines 650-666) – this was highlighted.

Yours sincerely,

Renata Szymanska

Faculty of Physics and Applied Computer Science

AGH University of Science and Technology

Reymonta 19, 30-059 Krakow, Poland

[email protected]

Reviewer 2 Report

The recent studies on the roles of tocopherols, β-carotene, and PQH2 in plants against oxidative stress have been reviewed. 

The sections are clearly demonstrated and organized. Most of the studies are cited in this review article. 

However, the typing errors and spelling need to be carefully checked. For example, Line 495: C. reinhardtii;  Line 509: as a consequence

Author Response

We carefully read the manuscript and correctedtyping errors and spelling. All changes were marked in red in the text.